# Hospital Inpatient Falls across Clinical Departments

**DOI:** 10.3390/ijerph18158167

**Published:** 2021-08-02

**Authors:** Marcin Mikos, Tomasz Banas, Aleksandra Czerw, Bartłomiej Banas, Łukasz Strzępek, Mateusz Curyło

**Affiliations:** 1Department of Bioinformatics and Public Health, Faculty of Medicine and Health Sciences, Andrzej Frycz Modrzewski Krakow University, 30-701 Krakow, Poland; mikos@ziz.com.pl; 2Department of Gynaecology and Oncology, Jagiellonian University Medical College, 31-501 Krakow, Poland; 3Department of Health Economics and Medical Law, Medical University of Warsaw, 02-091 Warsaw, Poland; ola_czerw@wp.pl; 4National Institute of Public Health NIH—National Research Institute, 00-791 Warsaw, Poland; 5Private Surgical Medical Practice, 31-261 Krakow, Poland; bartek14@wp.pl; 6Department of General Surgery, Regional Public Hospital in Bochnia, 32-700 Bochnia, Poland; strzepeklukasz@wp.pl; 7Orthopedic and Posttraumatic Rehabilitation Department, Medical University of Lodz, 90-419 Lodz, Poland; mateusz.curylo@azmmedical.pl

**Keywords:** fall assessment sheet, fall, elderly patients, hospitalization, risk management

## Abstract

Background: Inpatient falls are common hospital adverse events. We aimed to determine inpatient fall rates in an urban public hospital and analyzed their characteristics across clinical departments. Methods: The study was conducted in a 350-bed urban, multi-specialty public hospital in the 2013–2019 period. Patient data were retrieved from the hospital’s standardized falls reporting system. Descriptive statistics and statistical tests: chi2 and ANOVA tests with multiple comparison tests (post-hoc analysis) were used. For fall incidence estimation a joint-point regression was applied. *p*-value of 0.05 was considered as statistically significant for all the calculations. Results: The highest prevalence of falls was reported in the rehabilitation and internal medicine wards (1.915% and 1.181%, respectively), the lowest in the orthopedic (0.145%) and rheumatology wards (0.213%) (*p* < 0.001). The vast majority of falls took place in the late evening and during the night (56.711%) and were classified as bed falls (55.858%). The crude incidence rate (cIR) of falls was 6.484 per one thousand hospitalizations. In the 2013–2017 period, an increase in total cIR was observed, reaching the peak value in 2016; it was followed by a slight decline from 2017 to 2019, however, differences in changes were observed between the wards. Conclusion: Fall rates and trends as well as circumstances of inpatient falls varied significantly among clinical departments, probably due to differences in patient characteristics.

## 1. Introduction

The most widely used definition of a fall occurring within a healthcare setting is the one proposed by Nitz and Johnston, which describes it as “an unexpected event in which the participant comes to rest on the ground, floor, or lower level” [1]. Inpatient falls are the leading cause of hospital adverse events with incidence rate varying from 2.4 in large tertiary university hospitals to 9.1 in geriatric hospital departments per one thousand patient-days [1,2,3,4]. Two major types of fall risk factors were distinguished: (a) intrinsic factors comprising age, gender, musculoskeletal disorders, patient’s imbalance and using drugs; (b) extrinsic factors including the weaknesses of the health system in the medical equipment maintenance and design, human resources, communication, training, and team work [5].

Age > 85 years, the male sex, a recent fall, gait instability, agitation and/or confusion, new urinary incontinence or frequency, adverse drug reactions and neurological and cardiovascular instability are the predominant risk factors of inpatient falls [6,7]. Chronic diseases including diabetes and hypertension are also risk factors of falls and subsequent fractures [6,7,8]. Especially elderly patients are more likely to fall, due to balance and coordination deterioration, loss of skeletal muscles strength, as well as many other comorbidities associated with aging [6,7]. Physical activity, therefore, plays essential role in preventing falls. World Health Organization (WHO) recommendations emphasize the importance of physical activity in preventing falls especially among seniors and are consistent with studies showing that systematic movement exercises and balance training can lead to the alleviation of symptoms associated with balance disorders in the elderly, thus reduce the risk of falling [9,10]. Furthermore, 15.4% of patients experience a decline in mobility during hospital stay, particularly women with cognitive impairment and underweight are at high risk of reduced mobility, therefore physical rehabilitation during hospitalization is essential and was proved to reduce the risk of falls [11,12]. Finally, negative interactions between the intrinsic and extrinsic risk factors may lead to serious physical injuries [5].

Up to 42% of falls occur during walking (e.g., to the bathroom), while 7–14% take place during transferring (e.g., standing up, sitting down) or are bed-related (e.g., falling out of bed) [13,14,15,16]. Up to 80% of falls, however, occur when patients are not observed, as some patients initiate risky decisions concerning their mobility based on their own judgements, without asking health professionals for help [17,18].

Approximately 30–35% of falls occurring in healthcare facilities result in injury that can cost over USD 14,000 per incident adding, on average, 6.3 days to an individual’s length of stay [19]. Adverse outcomes associated with inpatient falls include bruises and fractures, depression and anxiety, prolonged lengths of stay, and even death [20]. Some fall-related incidents may even lead to a medical lawsuit; therefore, fall risk reduction via implementing monitoring and analyzing systems along with nursing care improvement and patients’ education have become one of the most important issues in medical safety.

In spite of the fact that most healthcare providers have implemented recommendations to identify patients at increased falls risk, and processes for collecting and reporting fall data, falls continue to occur [21]. Prevention of inpatient falls seems to be crucial for integrity of diagnostic and therapeutic processes. This can be achieved mainly by staff training, implementation of fall risk reduction programs and patient education. The first step in preventing falls is the identification of high-risk patients.

King et al. reported, however, unintended impact of fall prevention messages on nurses and older adult patients. Intense messaging from hospital administration to achieve zero falls resulted in nurses developing a fear of falls, protecting themselves and the unit, and restricting fall risk patients as a way to stop messages and meet the hospital goal [22]. Improperly, an adverse event is often considered as a synonym for “medical error,” “medical malpractice,” or “treatment failure.” However, the term “adverse event” also comprises treatment failures not directly caused by a healthcare provider, and not only by human medical errors [23]. Therefore, depenalization of unintended adverse effects, including inpatient falls, should be strongly considered. Furthermore, creating opportunities for anonymous recording of medical adverse events would significantly improve the number of reported cases. The goal of medical adverse event recording and reporting systems is primarily to identify possible risk factors in order to improve patients’ safety. Along with depenalization of unintended medical adverse events, a public insurance system should be established to cover justified patients’ claims [23,24].

In this study we aimed to investigate the occurrence of inpatient falls in all wards of a public city hospital in the past seven years, and to analyze the circumstances of these events; additionally, we investigated changes in the incidence of falls.

## 2. Materials and Methods

### 2.1. Setting and Sample

This study was conducted in a 350-bed specialist public hospital in the city of Krakow, Poland, after receiving Local Review Board consent. It included all adult inpatients from 1 January 2013 through 31 December 2019, in the clinical departments of internal medicine, rheumatology, rehabilitation, cardiology, neurology and orthopedics. No additional exclusion criteria were applied and all the records in the registry were completed with no missing data. The hospital is localized in a district where the percentage of people in the retirement age is higher than the city average [21]. Therefore, the mean of age in the sample is considerably high and exceeds the value of 75 years old (see Table 1).

Patient data were retrospectively retrieved from the hospital’s standardized falls reporting system. Fall report included the following data: (1) clinical department, (2) patient’s data, (3) time, (4) location, and (5) circumstances of the fall. In multiple fall cases, only the data of the first fall were analyzed. The hospital policy requires every hospital employee involved in any adverse event such as a patient’s fall to fill in a specific form immediately after the event. The form includes fields for entering all information mentioned above. It is then submitted to the Head of Department and its copy is sent to the office of the hospital director’s plenipotentiary for quality. The data are then added to the reporting system. The information is analyzed and appropriate corrective actions are taken.

### 2.2. Statistical Analysis

The Shapiro–Wilk test was used to determine the distribution of continuous variables. Variables that fit normal distribution were presented as mean values and standard deviation (SD), while those with distribution different from normal as medians and interquartile range (IQR). Analysis of variance (ANOVA) was used to compare more than two groups if variables fit normal distribution and Kruskal–Wallis ANOVA testing was performed if distributions were different from normal. Post-hoc tests were applied if appropriate. To compare the rough number of cases a chi-square test was chosen, and variables were presented as case numbers and percentage (%). The Neuman test was employed to evaluate if the trends of the annual number of admissions and the median hospital stays in the period 2013–2019 were significant. Additionally, Spearman correlation test was used to assess a possible relationship between the annual number of admissions and the annual number of inpatient falls as well as between the median length of hospital stay and the annual number of inpatients falls. Calculations were performed using STATISTICA data analysis software, version 12.0 (TIBCO Software Inc. (2017). Statistica (data analysis software system), version 13. Palo Alto, USA), and MedCalc Statistical Software, version 16.2.1 (MedCalc Software by Ostend, Belgium).

A join-point regression analysis using the Joinpoint Regression program, version 4.8.0.1 April 2020 (Information Management Services Inc., Rockville, MD, USA) was performed to determine the crude incidence rate of falls calculated as the number of falls per 1000 hospitalizations. The analysis included a logarithmic transformation of the rates, standard errors, and a maximum number of five join points with a minimum of 4 years between two join points [22]. The annual percentage change (APC) was subsequently calculated to quantify the trend over a fixed number of years as a geometric weighted average of the trend analysis. *p*-value of 0.05 was considered as statistically significant for all the calculations.

## 3. Results

In a seven-year period, there were 89,693 hospitalizations and 734 (0.818%) patients’ falls were reported. The study group comprised 325 (43.218%) males and 427 (56.782%) females. The mean age of patients who fell was 75.53 (±13.35) years with an average BMI of 27.08 (±5.39) kg/m^2^, and their median length of hospital stay was 16.5 (IQR: 7.0) days.

Additionally, in the rheumatology ward patients who fell were significantly younger compared to patients admitted to other wards (Table 1). Additionally, significant differences in female to male ratios were identified across the analyzed wards (Table 1). The highest rate of falls of female patients was observed in the rehabilitation ward; it was followed by the orthopedic and internal medicine wards. The lowest rate of falls of female patients was noted down in the neurology unit. There were no significant differences in BMI of fallers across the analyzed departments. Additionally, significant differences in female to male ratios were identified across the wards (Table 1). In the internal ward, the falls were predominately reported in the late evening and at night while in other wards they occurred mostly in the morning and in the afternoon (Table 1). Dementia was diagnosed in 1 in 10 of fallers from the cardiology, internal and neurology wards, while disorientation was recognized in 17% of patients who fell in the neurology and cardiology wards followed by 12% of patients in the internal medicine ward, and the differences between the wards were significant (Table 1). Bed falls were typical for all these wards; however, a rehabilitation and rheumatology corridor was the second most common location where falls occurred contrary to other wards where bathroom falls were more common (Table 1).

The longest hospital stay was reported in the rehabilitation ward followed by the internal medicine and orthopedics wards, while the shortest stay was in the rheumatology ward and the highest number of annual admissions was to the internal medicine ward while the lowest to the rehabilitation department (Table 1).

In the cardiology department both the trends of the annual admissions and the median length of hospital stay were insignificant (Figure 1). Additionally, no association between the annual number of admissions and the annual number of inpatient falls or a relationship between the median length of hospital stay and the annual number of inpatients falls were proven. Contrary to this, in the internal medicine ward the annual admissions trend increased significantly while the trend of median hospital stay was insignificant (Figure 3). There was also a significant positive association between the annual number of admissions and the annual number of inpatient falls (R = 0.775; *p* = 0.041) while no correlation between the median length of hospital stay and the annual number of inpatient falls was observed. In the neurology department both the trends of annual admissions and the median length of hospital stay increased significantly (Figure 3). There was a significant positive correlation between the annual number of admissions, the median length of hospital stay and the annual number of inpatient falls (R = 0.982; *p* = <0.001 and R = −0.908; *p* = 0.004, subsequently). The annual admissions trend in the orthopedics department increased significantly while the trend of the median hospital stay was insignificant (Figure 3). There was no correlation between the annual number of admissions and the annual number of inpatient falls or between the median length of hospital stay and the annual number of inpatients falls. In the rheumatology department both trends were significant, however, the annual admissions trend was increasing, while the trend of the median hospital stay decreased. Additionally, no correlations between the annual number of admissions and the annual number of inpatient falls or between the median length of hospital stay and the annual number of inpatient falls were found. In the rehabilitation ward only the trend of the median hospital stay decreased significantly, while the annual admissions trend was insignificant. Additionally, there were no correlations neither between the annual number of admissions and the annual number of inpatient falls nor between the median length of hospital stay and the annual number of inpatients.

The crude incidence rate (cIR) of falls was 6.484 per one thousand hospitalizations. In the 2013–2017 period, an increasing trend of total cIR was observed, reaching the peak value in 2016; it was followed by a slight decline from 2017 to 2019 (Figure 2).

Different changes in fall incidence were observed in the analyzed wards. The highest incidence of falls was reported in the rehabilitation ward, where the maximum cIR of 41.06 was noted down while its lowest level for that ward was 2.53 (Figure 3). The second highest cIR of 16.94 falls was observed in the internal medicine ward; here, its lowest value was 2.11 (Figure 3). The lowest cIR of falls was observed in the neurology ward, and it was in the 0.09–1.33 range (Figure 3). In the orthopedics ward, a trend analysis was unavailable due to lack of reported cases in the 2013–2015 and 2018–2019 periods.

In the cardiology department, after a gradual increase in the number of falls from 2013 to 2016, we observe a rapid acceleration of this trend that reached the peak of cIR at the level of 14.11 in 2017; it was followed by a significant decline in the 2017–2019 period (Figure 3). Similarly to the cardiology unit, also in the rehabilitation ward a two-part trend of patient falls was observed. After an initial enormous increase of cIR from 2.53 to 41.06 in the 2013–2017 period, a steep decrease can be seen from 2017 to 2019, with significant APC (Figure 3). In the internal medicine ward, a rapid upward trend was observed from 2013 to 2015, with a significant increase of falls. From the year 2015, however, this trend stabilized, with an insignificant APC, reaching the peak cIR of 16.94 in 2019 (Figure 3).

Contrary to what was mentioned above, we observed a continuous increase of inpatient falls in the neurology and the rheumatology departments, although cIR of falls differed significantly between these two wards (Figure 3). In the neurology ward, after a gentle increase in falls, an acceleration of the trend was observed in the 2013–2017 period; it reached the highest cIR value of 18.48 in the year 2019 and showed a significant APC for the whole analyzed period (Figure 3). In the rheumatology ward, although an upward trend was observed for the entire analyzed period from the years 2013–2019, the reported APC was insignificant and showed the lowest values of cIR of falls as described above (Figure 3). As we have already mentioned, a trend analysis was unavailable for the orthopedics ward.

## 4. Discussion

Our results are consistent with findings of Healey et al., who reported rates of falls per 1000 bed days between 2.1 and 8.4, depending on the hospital profile, and significantly lower than fall rates presented by Schwendimann and colleagues, who showed that 7.2% of hospitalized patients experienced falls [16,22]. Similarly to our findings, they reported significant differences in characteristics of fallers and circumstances of falls; however, they investigated internal medicine, surgical and geriatric departments [22,23]. These significant differences considered patients’ age, length of hospital stay as well as comorbidities and the circumstances of falls. They can be easily explained by the fact that in distinct departments patients are diagnosed with different health problems and have individualized treatment. Results presented by Tayabe, however, showed that one fourth of recorded falls were not registered in the incident reporting systems [24]. It is a well-known fact that only a part of the incidents occurring in a hospital is recognized in a voluntary incident reporting systems [25]. Rates of falls recorded based on incident reports, vary remarkably between hospitals and this inconsistency in the rate of falls may be the result of reporting bias of medical staff [25]. Consistently, Healey et al. confirmed that that the rate of falls in acute hospitals varied remarkably between hospitals from 0.2 to 11.5 per 1000 bed days [26]. Reporting bias is a serious problem especially when the precise incidence and detailed information on incidents are required. Epidemiological study of inpatient falls, validation of countermeasures against falls, and development of risk assessment systems for inpatient falls can be effective only if based on truth and verified.

According to our result, the highest fall risk was in the rehabilitation ward followed by internal medicine department, while orthopedics, cardiology and rheumatology patients were at the lowest risk of falls. This knowledge is essential to improve inpatient fall prevention; however, not only a common fall risk in each ward should be evaluated but also an assessment of individual fall risk must be conducted on admission of every patient. There is, however, no consistent evidence that interventions to prevent falls among hospital inpatients are effective [9,10], although many of the published studies were underpowered or methodologically flawed.

Basically, the risk of inpatient falls is positively correlated with the length of hospital stay and rises significantly from the 11th day of hospitalization [27]. Similarly, in intensive care units the risk of inpatient falls increases 9.9 times if the hospitalization exceeds 19 days [28]. A positive and significant correlation between the risk of inpatient falls and hospitalization length was also confirmed in palliative care units [29]. Furthermore, frequent rotations of nursing staff and extensive workload resulted in omitting many important procedures and activities that are directly relevant to patient safety, such as lack of care planning (18.9%), lack of updating medical records (21.7%) and reducing nursing care (23.9%) [29,30,31,32].

In the investigated population increases in the incidence of inpatient falls on neurology and rheumatology wards were observed in conjunction with a decrease in the median length of hospital stay and a rise of new admissions.

To the best of our knowledge a very large number of papers evaluated inpatients falls risks but only few studies evaluated changes in the incidence of falls. The novelty of our study is the evaluation of incidence across different clinical departments. In most departments we observed a decrease in the incidence of falls during the last three or five years, except for the internal medicine ward, where this trend was stable. The neurology ward was the only one showing a significant increase in the incidence of inpatient falls. A trend analysis allows to predict potential changes in fall incidence in the future and such information is essential for proper planning of fall prevention activities that should be tailored for each department separately.

We are also aware that our study has some limitations that must be discussed. The major drawback of this data set is lack of description of what medications associated with the risk of falls were being used by patients in which falls occurred. However, the lack of this analysis does not discredit the results. Secondly, due to the huge number of staff involved in reporting inpatient falls, as well as staff fluctuation during the study period, the quality of data on registered patient falls may vary. Thirdly, falls risk factors were unavailable for the analysis. Finally, we used data from just one hospital which provides treatment for a specific population with the percentage of people in the retirement age higher than the city average. These limitations, however, did not prevent us from achieving the aim of the study and presenting reliable results. The strength of this study is that data were available from patients treated in different clinical departments in a tax-funded healthcare system in public hospital. Furthermore, a comprehensive analysis of inpatient falls in a large sample across clinical departments is the major power of this study. The subject clearly needs meta-analysis based on data from different hospitals functioning in various surroundings, which could provide a more reliable estimate of fall rates. Both characteristics of hospitals and patients treated could be analyzed as potential moderators accounting for differences between results based on separate datasets.

## 5. Conclusions

Inpatient falls remain the leading cause of adverse events in hospitals. According to the presented results, the prevalence of falls was equal to 0.82%. However, there are significant differences in the incidence of inpatient falls between different wards. The highest fall risk was in the rehabilitation ward followed by internal medicine department, while orthopedics, cardiology and rheumatology patients were at the lowest risk of falls. The falls occurred most frequently between 24:00 and 6:00 and were more prevalent in the group of female patients. Furthermore, in different clinical departments, distinct changes in the incidence of inpatient falls were reported. In our opinion, in order to improve patient safety, not only reporting rough numbers of inpatient falls but also an analysis of changes seem to be crucial, as only this allows to predict potential future changes in falls, which is essential for proper planning of fall prevention activity.

## Figures and Tables

**Figure 1 ijerph-18-08167-f001:**
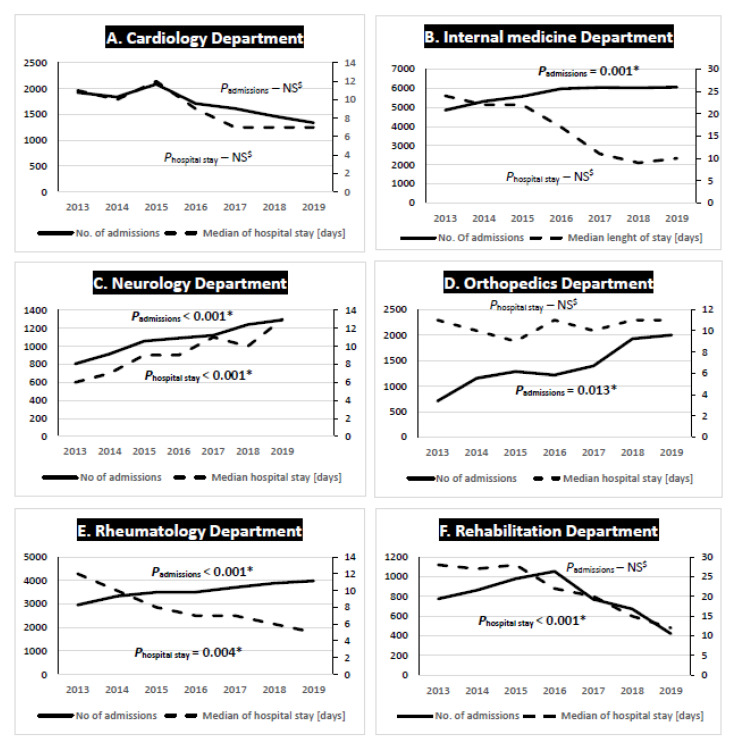
Trends of median hospital stay and annual number of admissions across the analyzed clinical departments. ^$^ NS—non-significant; * significant *p*-value.

**Figure 2 ijerph-18-08167-f002:**
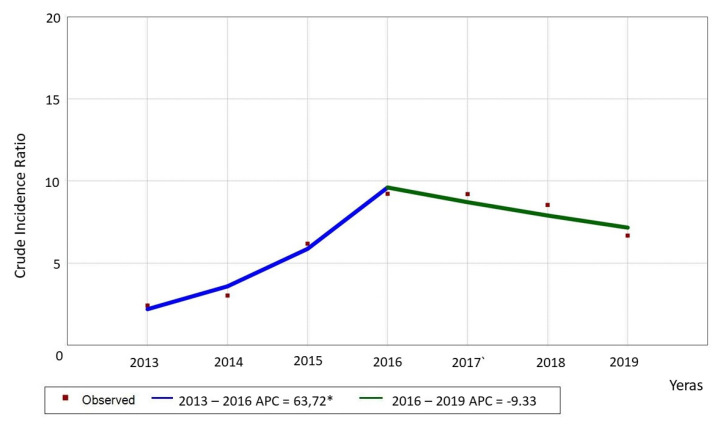
Trend of patient falls (Crude incidence ratio [cIR] per 1000 patients-beds) for all the departments. * please pay attention to a different scale.

**Figure 3 ijerph-18-08167-f003:**
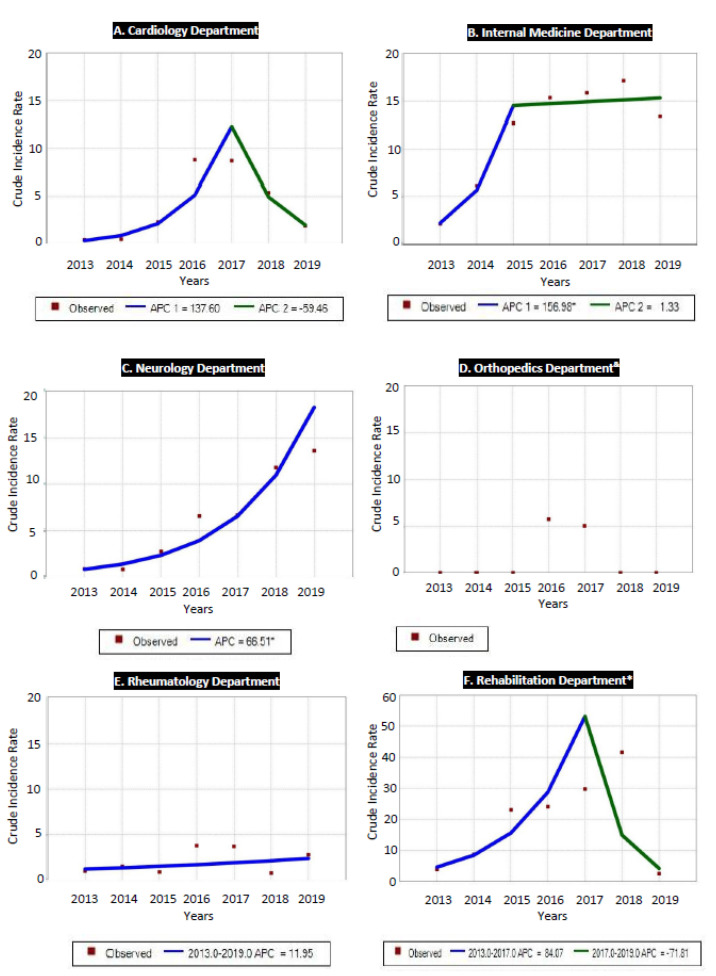
Trends of falls (crude incidence ratio (cIR) per 1000 patient beds) across the analyzed clinical departments; & no trend available due to 0 cases of inpatient falls in several years; * please pay attention to a different scale.

**Table 1 ijerph-18-08167-t001:** Characteristics of fallers and circumstances of falls.

Departments	Cardiology	Internal Medicine	Neurology	Orthopedics	Rheumatology	Rehabilitation	*p*
	A	B	C	D	E	F	
Total number of hospitalizations (N)	11980	39811	7517	9673	24849	5536	<0.001 ^$^
Total number of falls (%)	47 (0.39%)	469 (1.18%)	45 (0.60%)	14 (0.15%)	53 (0.21%)	106 (1.92%)	
Females N (%)	18 (38.30%)	256 (54.58%)	17 (37.78%)	8 (57.14%)	43 (81.13%)	81 (76.42%)	A vs. B < 0.008 ^$^ A vs. E = 0.004 ^$^ A vs. F = 0.008 ^$^ C vs. E < 0.001 ^$^ C vs. F < 0.001 ^$^
Males N (%)	29 (61.7%)	213 (45.42%)	28 (62.22%)	6 (42.90%)	10 (18.90%)	25 (23.58%)	
Age (years)	77.94	77.70	77.76	71.29	65.81	72.81	B vs. A < 0.001 ^$^ B vs. C < 0.001 ^$^ B vs. E = 0.004 ^$^ C vs. E = 0.008 ^$^
mean (±SD *; range)	(±10.96; 36)	(±12.08; 73)	(±11.17; 64)	(±8.46; 27)	(±19.07; 63)	(±11.51; 59)	
BMI (kg/m^2^)	26.57	27.06	26.43	26.24	26.95	28.31	0.083
mean (±SD *)	(±5.01)	(±5.58)	(±4.87)	(±2.89)	(±4.62)	(±5.64)	
Length of stay (days)	9	14	9	10	7	22	F vs. A < 0.001 ^$^ F vs. B < 0.001 ^$^ F vs. C = 0.004 ^$^ F vs. D = 0.003 ^$^ F vs. E < 0.001 ^$^ A vs. E < 0.001 ^$^ A vs. B = 0.036 ^$^ B vs. E < 0.001 ^$^
Median; IQR **	IQR **: 3.5	IQR **: 4.0	IQR **: 6.5	IQR **: 5.5	IQR **: 3.0	IQR **: 11.5	
Time of fall							0.018 ^$^
6:00-12:00	12 (25.53%)	89 (19.98%)	12 (26.67%)	6 (42.86%)	14 (26.42%)	21 (19.81%)	
12:00-18:00	15 (31.92%)	91 (19.40%)	12 (26.67%)	0 (0.00%)	17 (32.08%)	29 (27.36%)	
18:00-24:00	11 (23.40%)	123 (26.23%)	11 (24.44%)	4 (28.57%)	9 (16.98%)	31 (29.25%)	
24:00-6:00	9 (19.149%)	166 (35.39%)	10 (22.22%)	4 (28.57%)	13 (24.53%)	25 (23.59%)	
Patient conditions							B vs. E = 0.006 ^$^ C vs. E = 0.012 ^$^
Dementia	5 (10.64%)	56 (11.940%)	5 (11.11%)	0 (0.000%)	1 (1.89%)	2 (1.89%)	
Disorientation	8 (17.02%)	59 (12.58%)	8 (17.78%)	0 (0.000%)	0 (0.00%)	5 (4.72%)	
Psychomotor disorders	2 (4.26%)	19 (4.05%)	2 (4.44%)	3 (21.43%)	0 (0.00%)	4 (3.77%)	
Loss of consciousness	0 (0.00%)	6 (1.28%)	0 (0.000%)	0 (0.00%)	3 (5.66%)	0 (0.00%)	
None	32 (70.15%)	329 (66.67%)	30 (66.67%)	11 (78.57%)	49 (92.45%)	95 (89.62%)	
Place of fall							C vs. E < 0.001 ^$^ D vs. E = 0.011 ^$^
Bed	30 (59.92%)	281 (59.92%)	28 (62.22%)	8 (57.14%)	16 (30.19%)	47 (44.34%)	
Bathroom	11 (22.81%)	107 (22.81%)	11 (24.44%)	0 (0.00%)	14 (26.42%)	19 (17.93%)	
Corridor	6 (12.77%)	81 (17.27%)	6 (13.33%)	6 (42.86%)	23 (43.40%)	40 (37.74%)	

* SD—standard deviation; ** IQR—interquartile range; ^$^
*p* statistically significant. Kruskal–Wallis ANOVA with post hoc multiple cooperation of mean ranks.

## Data Availability

The data presented in this study are available on request from the corresponding author. The data are not publicly available due to restriction.

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
