# Peer review of "Hospital Inpatient Falls across Clinical Departments"

_ijerph, 2021, doi:10.3390/ijerph18158167_

Round 1

Reviewer 1 Report

It´s a good paper, however you should improve some parts of this manuscript.

INTRODUCTION

It would be interesting that you add, some information about the importante of Physical Activity for to improve and to prevent fall risk, and the importance of cognitive impairment to suffer a fall, this issue is a novelty.

METHODS

Have you realized the statistical analysis adjusting for potential confounders? 

RESULTS

In table 1, you should improve the title Rheumatology, you should improve.

Author Response

Comments and Suggestions for Authors

It´s a good paper, however you should improve some parts of this manuscript.

@ Thank you for the positive feedback on our manuscript. 

INTRODUCTION

It would be interesting that you add, some information about the importante of Physical Activity for to improve and to prevent fall risk, and the importance of cognitive impairment to suffer a fall, this issue is a novelty.

@ We agree – the following chapter on physical activity in falls prevention was added to the Introduction:

“Physical activity, therefore, plays essential role in preventing falls. World Health Organisation (WHO) recommendations emphasize the importance of physical activity in preventing falls especially among seniors and are consistent with studies showing that systematic movement exercises and balance training can lead to the alleviation of symptoms associated with balance disorders in the elderly, thus  reducethe risk of falling [9,10]. Furthermore, 15.4% of patients experience a decline in mobility during hospital stay, particularly women with cognitive impairment and underweight are at high risk of reduced mobility, therefore physical rehabilitation during hospitalisation is essential and was proved to reduce the risk of falls [11,12]. Finally negative interactions between the intrinsic and extrinsic risk factors may lead to serious physical injuries [5].”

METHODS

Have you realized the statistical analysis adjusting for potential confounders? 

@ Thank you for this comment. Potential confounders including not reporting of falls were discussed in “Limitation of the study” Reporting of falls that did not occurred was ruled out.

 RESULTS

In table 1, you should improve the title Rheumatology, you should improve.

@ Thank you for this remark -  corrected as indicated.

Reviewer 2 Report

This manuscript reports a seven-year observational survey of patient fall incidents in a public hospital.  The event data were collected across six department wards and compared.  The authors also analyzed the year-by-year trends of the crude incidence rates in the departments and identified difference in the trends among the departments.  Although the long-term data collection provides a reliable evidence of the incident rate, the interpretation of the result presented in the manuscript includes a serious flaw.  The reviewer do not recommend publication of this manuscript in the current form. 

-- As introduced in the introduction and discussion sections, there are already a number of studies that reported the fall incidence rates in the similar settings.  The novelty of this study, therefore, should lie in the evaluation of the trends of cIRs across the different departments.  However, the authors fails to clarify the origins of the observed trends, that is, what caused the annual change in cIR in each department.  The fluctuations of the cIR trends can be attributed merely to artifacts relating to the procedure of incident reports because the data were collected in one hospital. 

-- The data presentation in Table 1 has some issues.  The percentage values have too many decimal figures.  The statistical methods used to obtain P values should be declared.  The results of post-hoc tests should also be displayed here.

Author Response

Dół formularza

Rev 2

Open Review

Comments and Suggestions for Authors

This manuscript reports a seven-year observational survey of patient fall incidents in a public hospital.  The event data were collected across six department wards and compared.  The authors also analyzed the year-by-year trends of the crude incidence rates in the departments and identified difference in the trends among the departments.  Although the long-term data collection provides a reliable evidence of the incident rate, the interpretation of the result presented in the manuscript includes a serious flaw.  The reviewer do not recommend publication of this manuscript in the current form. 

@ We acknowledge this serious criticism.

-- As introduced in the introduction and discussion sections, there are already a number of studies that reported the fall incidence rates in the similar settings.  The novelty of this study, therefore, should lie in the evaluation of the trends of cIRs across the different departments.  However, the authors fails to clarify the origins of the observed trends, that is, what caused the annual change in cIR in each department.  The fluctuations of the cIR trends can be attributed merely to artifacts relating to the procedure of incident reports because the data were collected in one hospital. 

@ We analysed trends in median hospital stay and annual number of admissions in every ward – the results were presented in Figure 3 and in the Results section. Additionally discussion was deepened as following:  

The risk of inpatients falls is positively correlated with the length of hospital stay and rises significantly after from the 11th day of hospitalisation [1]. Similarly in intensive care units risk of inpatients falls increases 9,9 times if the hospitalisation exceeds 19 days [2]. Positive and significant correlation between risk of inpatients falls and hospitalisation length was also confirmed in palliative care units [3]. Furthermore, frequent rotations of nursing staff and extensive workload, resulted in omitting many important procedures and activities that are directly relevant to patient safety, such as lack of care planning (18.9%), lack of updating medical records (21.7 %) and reducing nursing care (23.9%).

In investigated population increases in incidence of inpatients falls on neurology and rheumatology wards were observed in conjunction with decrease with median length of hospital stay and a rise of new admissions.

-- The data presentation in Table 1 has some issues.  The percentage values have too many decimal figures.  The statistical methods used to obtain P values should be declared.  The results of post-hoc tests should also be displayed here.

@ Thank you for this comment – Table 1 was changed as indicated.

Reviewer 3 Report

Dear authors:

The article does not bring new knowledge, but it contributes to the discussion of the topic with the international literature, so I consider that being improved it can be published.
The introduction explores the literature review of the topic and makes it possible to justify the relevance of the study.
The objective of the abstract and the introduction are slightly different and should be standardized.
I do not agree that this study allows investigating trends in the incidence of falls and therefore I suggest changing the title of the article and removing this objective from the introduction.

The method needs to be improved by introducing the study design, clarifying when data is collected, how many investigators are involved, what are the procedures in data collection to ensure accuracy. I recommend that you consult the reporting guideline (STROBE -Strengthening the Reporting of Observational Studies in Epidemiology), and follow the structure.

There is mention, in the method, that the study population was adults, but the average ages presented in the results refer to an elderly population. It needs to be clarified whether they included adults and elderly or only elderly.

In the results delete item 3.2 - Figures, Tables and schemes whose reference is included in the body text.
The discussion can be deepened.
The limitations of the study should be presented.
The conclusion should be improved by presenting a summary of the results - prevalence of falls, occurrence in which service, during which period of the day, in which gender and ages.

Author Response

Rev 3

Open Review

Comments and Suggestions for Authors

Dear authors:

The article does not bring new knowledge, but it contributes to the discussion of the topic with the international literature, so I consider that being improved it can be published.

@ Thank you for this comment.

The introduction explores the literature review of the topic and makes it possible to justify the relevance of the study.

@ Thank you for the positive feedback.

The objective of the abstract and the introduction are slightly different and should be standardized.
I do not agree that this study allows investigating trends in the incidence of falls and therefore I suggest changing the title of the article and removing this objective from the introduction.

@ We deleted investigating trends in the incidence of falls from the title, the abstract and the study objective.

The method needs to be improved by introducing the study design, clarifying when data is collected, how many investigators are involved, what are the procedures in data collection to ensure accuracy. I recommend that you consult the reporting guideline (STROBE -Strengthening the Reporting of Observational Studies in Epidemiology), and follow the structure.

@ We added a description of data collection which is based on the hospital policy.

There is mention, in the method, that the study population was adults, but the average ages presented in the results refer to an elderly population. It needs to be clarified whether they included adults and elderly or only elderly.

@ We completed the sample characteristics to explain high mean value of the patients’ age.

In the results delete item 3.2 - Figures, Tables and schemes whose reference is included in the body text.

@ We altered the body of text and deleted the Total column in Table 1 to avoid repetitions.

The discussion can be deepened.

The limitations of the study should be presented.

@ We broadened discussion by completing the list of limitations and proposing possible idea for future research.

The conclusion should be improved by presenting a summary of the results - prevalence of falls, occurrence in which service, during which period of the day, in which gender and ages.

@ We improved conclusion by presenting a summary of the results regarding prevalence of falls, occurrence in which service, period of the day and gender.

Reviewer 4 Report

In the present manuscript, the authors investigated the inpatients falls across clinical departments. In my opinion, the study is well described and interesting, but some improvements are necessary.

Materials and methods

Study populations: there was some exclusion criteria? Moreover, was assessed previous falls before hospitalization?

Population description: the authors reported also mean age. What is the range?

Results: if possible, can the authors analysed the data separately between fallers and no fallers before hospitalization? If not, this could be included as study limitation.

Author Response

Rev 4

Open Review

Comments and Suggestions for Authors

In the present manuscript, the authors investigated the inpatients falls across clinical departments. In my opinion, the study is well described and interesting, but some improvements are necessary.

@ Thank you for the positive feedback on our manuscript. 

Materials and methods

Study populations: there was some exclusion criteria? Moreover, was assessed previous falls before hospitalization?

@ Than you for this comment. No exclusion criteria were applied – all records in the registry were completed therefore no data were missing – all reported falls were included in the analysis – Material and Methods section was supplemented with this information.

Population description: the authors reported also mean age. What is the range?

@ The range of age in the whole study group was 24-97 years – this information was supplemented in the results. Additionally range of age was added to Table 1

Results: if possible, can the authors analysed the data separately between fallers and no fallers before

@ Thank you for this valuable remark, unfortunately Inpatients falls registry includes only information concerning patients who fall during hospital stay and does not provide information concerning previous falls -  that’s why separate analysis between faller and no fallers is unavailable in this project. This can be analysed based on fall-risk assessment done on admission – we therefore plan do design and conduct relevant prospective study.

Round 2

Reviewer 2 Report

The authors added the trends of median hospital stay and annual numbers of admissions across the analyzed clinical departments to the results.  In discussion, the authors also introduced the previous studies that reported the relations between the cIR trends and these trends.  However, the authors do not show any results of statistical analysis that support any relation between the trends of cIR (Fig. 2) and added data (Fig 3).  Therefore, the authors fail to solve the issue raised by the reviewer. 

Author Response

@ In this critic review, unfortunately no clues were given haw this important issue can be solved, therefore we applied Neuman test to evaluate trends of annual number of admissions and median hospital stays in the period 2013-2019 were significant, additionally Spearman correlation was employed to  assess possible relationship between annual number of admissions and annual number of inpatients falls and  between median length of hospital stay and annual number of inpatients falls, The following results were obtained:

Cardiology Department: both the trends of annual admissions to the ward and median length of hospital stay were insignificant (Fig 3). Additionally no association between annual number of admissions and annual number of inpatients falls neither  relationship between median length of hospital stay and annual number of inpatients falls were proven.

Internal Medicine Department: The annual admissions trend increased significantly while the trend of median hospital stay was insignificant (Fig 3). There was a significant positive association between annual number of admissions and annual number of inpatients falls (R=0,775; p=0,041) while no correlation between the median length of the hospital stay and the annual number of inpatients falls was found.

Neurology Department: both the trends of annual admissions to the word and median length of hospital stay increased significantly (Fig 3). There was a significant positive correlation between the both: the annual number of admissions, the median length of hospital stay and the annual number of inpatients falls (R=0,982; p=<0,001 and R=-0,908; p=0,004, subsequently).

Orthopaedics Department: The annual admission trend increased significantly while the trend of the median hospital stay was insignificant (Fig. 3). There was no correlation between the annual number of admissions and the annual number of inpatients falls neither between the median length of hospital stay and the annual number of inpatients falls.

Rheumatology department: Both trends were significant, however the annual admission trend was increasing, while the trend of median hospital stay decreased. Additionally no correlations between the annual number of admissions and the annual number of inpatients falls neither between the median length of hospital stay and the annual number of inpatients falls were found

Rehabilitation Department: Only the trend of median hospital stay decreased significantly, while the number of annual admission trend was unsignificant. Additionally there were no correlations between the annual number of admissions and the annual number of inpatients falls neither between the median length of hospital stay and the annual number of inpatients.

Subsequently sections: Materials and Methods, Results and Figure 3 were supplemented with relevant information.

We strongly believe that performed analysis address directly the reviewer`s concerns.

Reviewer 3 Report

The authors made the changes which improved the overall quality of the article.

Congratulations on the research that I consider to be fit for publication.

Author Response

@ Thank you for the positive feedback on our manuscript. 

Reviewer 4 Report

The authors answer adequately the request.

Author Response

(The authors gave the same response as above.)

Round 3

Reviewer 2 Report

The authors added the results of the statistical analysis on the relations between the cIR trends and the trends of annual admissions and the median length of hospital stay, and clarified significant correlations in some departments.  This revision successfully addressed the concern of the reviewer, and now the publication of this manuscript can be recommended.

This manuscript is a resubmission of an earlier submission. The following is a list of the peer review reports and author responses from that submission.